# The Discharge Performance of Mg-3In-xCa Alloy Anodes for Mg–Air Batteries

Huikun Liu [1,†], Guochen Zhao [1,*,†], Hang Li [1,*], Shouqiu Tang [1], Dapeng Xiu [1], Jin Wang [1], Huan Yu [1], Kaiming Cheng [1], Yuanfeng Huang [2] and Jixue Zhou [1,*]

1 Shandong Provincial Key Laboratory of High Strength Lightweight Metallic Materials, Advanced Materials Institute, Qilu University of Technology (Shandong Academy of Sciences), Jinan 250014, China; liuhuikun2022@163.com (H.L.); tangshq@sdas.org (S.T.); xiudp@sdas.org (D.X.); wangjin@sdas.org (J.W.); yuhuan@sdas.org (H.Y.); chengkm@sdas.org (K.C.)

2 Shandong Giant E-Tech Co., Ltd., Jinan 250000, China; huangyf@sdtj.sd.cn

\* Correspondence: zhaogch@sdas.org (G.Z.); lih@sdas.org (H.L.); zhoujx@sdas.org (J.Z.)

† These authors contributed equally to this work.

**Abstract:** Considering the advantage of safety and cost, designing a Mg–air battery with high capacity has been highly sought in recent years. However, self-corrosion and passivation of Mg anode critically reduce discharge performance, hindering the large-scale application of Mg–air batteries. In this study, a series of as-cast and extruded Mg-3In-xCa alloys were successfully fabricated. Microstructures, chemical composition, and discharge performance were investigated to optimize the content of Ca (x). The selected Mg–air battery with Mg-3In-3Ca alloy as anode represented the best battery performance, including 0.738 V of discharge voltage, 1323.92 mAh g$^{-1}$ of specific capacity, and 61.74% of anodic efficiency at discharge current density of 30 mA cm$^{-2}$. All of its parameters were vastly superior to pure Mg–air battery. In addition, the synergistic effects of In and Ca on promoting electrode properties were evaluated in detail, using SEM and electrochemical analysis, which is expected to trigger follow-up research in designing high-performance Mg–air batteries.

**Keywords:** Mg–air battery; alloying; Mg anode alloy; electrochemistry; discharge performance

## 1. Introduction

With high theoretical voltage (3.09 V), high theoretical specific capacity (2205 mAh g$^{-1}$), and high energy density (6800 Wh kg$^{-1}$), magnesium (Mg) has been widely investigated in recent years as an electrode with a large capacity. One of the Mg-based batteries, the Mg–air battery, received much more attention, resulting from its theoretical electrochemical performance, as well as its large abundance, lower cost, and safer performance than organic system batteries. However, hydrogen evolution corrosion and grain shedding, which occur during the discharging process of Mg anode, reduce the anode utilization of Mg–air batteries. In addition, the passivation films attached compactly on the surface of the Mg anode limit the discharge reaction activity of the Mg anode during the discharging process [1,2]. Highly active and low self-corrosive Mg anodes designed by alloying and processing are therefore highly sought.

In reference to previous studies, an effective solution to settle the above problems is to alloy Mg anode, so as to reduce the self-corrosion rate of Mg anode and synchronously induce the formation of loose and porous magnesium oxide film, which results in the film's tendency to fall off from the anode surface rather than attach to it [3–5]. Commonly used alloy elements are Al, Li, Pb, Sn, Ga, and In. Some researchers also study the influence of rare-earth elements on the discharge activity of Mg anode in order to obtain quality Mg–air batteries. According to a paper by Wu et al. [6], Mg-Hg-Ga alloy exhibited excellent corrosion resistance, which dramatically improved the utilization efficiency of Mg anode, up to 88.94 ± 0.33%, as was also reported by Wang et al. in their study of Mg-Al-Pb-Ce-Y

alloy [5]. The second phase formed between alloy elements and magnesium matrix can be used as a barrier to hinder the corrosion of magnesium alloy, reduce the self-corrosion of battery during discharge, and improve the anode utilization rate of the battery. However, elements such as Hg, Pb, Ga, etc. are environmentally harmful, which inevitably cause hazardous conditions across the globe. Another strategy is regulating the microstructure of alloy anode. Parameters such as grain size, distribution, grain boundaries, and the corresponding contents have been demonstrated to affect the electrochemical performance of Mg anode. Such regulation could mainly be realized by heat treatment and plastic deformation. According to the research of Xiong et al. [7], extrusion makes the second phase of Mg-Al-Sn alloy redistribute, improving the discharge performance of Mg-Al-Sn alloy anode. This coincides with the extruded Mg-Al-Sn-Mn alloy reported by Zheng et al. [8]. As for grain size, grain refinement is considered to effectively enhance the corrosion resistance of Mg alloy; for example, Li et al. [9] demonstrated that the extrusion refined the grain size of Mg-Al-In alloy and improved the anode utilization efficiency.

Recently, element indium (In) has received much attention because it improves the discharge activity of the Mg anode and accelerates the shedding of corrosion products. The improvement of the electrode performance by adding indium benefits from the destructive effect on the compactness of the passivation film and the inhibiting effect on self-corrosion. In addition, element calcium (Ca) has more negative standard electrode potential than Mg [3,10]; therefore, adding Ca into Mg is expected to improve the discharge voltage of Mg-based anode. It is reported that adding Ca into Mg-Bi alloy could thin the discharge product layer and enlarge its internal stress to produce numerous cracks, thus activating anode [11]. According to the research of Yuasa et al. [12], after adding Ca into the Mg-Al-Mn alloy anode, the thickness and compactness of the discharge products layer were reduced, which improved the discharge activity of the anode. Moreover, Ca refines the grain size, contributing to increasing the utilization efficiency of the anode. In the second phase, $Mg_2Ca$ precipitated along the grain boundaries, stably acting as a corrosion barrier and increasing charge transfer resistance, which helps to enhance the corrosion resistance of the Mg anode [13].

In our previous study, we explored the effects of In content on the microstructure and discharge performance of Mg-In alloy. The discharge curve showed, at a discharge current of 30 mA/$cm^2$, Mg-3In acquired more negative electrode potential and higher discharge-voltage capacity density than pure Mg and other Mg-xIn alloys, which demonstrated that 3% In is beneficial for activating Mg-based anodes. However, the utilization efficiency of Mg-In alloy anode was relatively low, only 46.14%. The purpose of this study was to experimentally investigate the effect of adding Ca into Mg-3In alloy anodes with different manufacturing processes on the performance of Mg–air batteries and optimize Ca content and the manufacturing process of Mg-3In-xCa alloys to acquire a Mg alloy anode material with high performance for Mg–air batteries. In this study, we investigated adding Ca into Mg-3In alloy, in order to enhance the corrosion resistance of anode and promote anodic efficiency. The extrusion process was also applied to regulate microstructure. A series of activated and inhibited Mg-3In-xCa anodes were successfully fabricated and discussed for future application in high-performance Mg–air batteries.

## 2. Materials and Methods

### 2.1. Materials Preparation

Pure Mg (>99.99%), high-purity indium particles (>99.999%) and Mg-25 wt.% Ca intermediate alloys were used.

### 2.2. Fabrication of Anode Alloys

Under the protection of carbon dioxide and sulfur hexafluoride mixture gas, the metal materials were melted in a low carbon steel crucible at 720 ± 20 °C and stirred for 30 s. After that, the temperature was increased to 730 ± 20 °C, a GJLJ17080303-1 refiner was added, and the mixture was continuously stirred for 30 s. Finally, the temperature was reduced to

690 ± 20 °C for water quenching. The as-cast specimens were solution-treated at 400 °C for 24 h and quenched in water. The solution-treated specimens were further processed with hot extruding at 380 °C with an extrusion ratio of 30:1. A bar with a diameter of 23 mm was finally obtained. The actual composition of as-cast alloy was analyzed by an energy-dispersive spectrometer (EDS), and the results are listed in Table 1.

**Table 1.** Composition of casting alloy.

| Samples | Mg (wt.%) | In (wt.%) | Ca (wt.%) |
|---------|-----------|-----------|-----------|
| IC31 | 95.1 | 3.36 | 1.54 |
| IC32 | 94.51 | 3.35 | 2.14 |
| IC33 | 92.75 | 3.80 | 3.45 |
| IC34 | 91.92 | 3.76 | 4.32 |
| IC35 | 91.4 | 3.25 | 5.35 |

### 2.3. Microstructure Characterization

The samples were mechanically ground with SiC sandpapers up to 2000 grit (240-600-800-1200-2000) and polished with diamond polishing paste to remove scratches. Afterward, they were etched by nitric acid acetic acid solution (1% nitric acid, 20% acetic acid, 60% ethylene glycol, and 19% water). Surface microstructures of the samples were characterized by an optical microscope (OM, ZEISS Axio observerAlm) and a scanning electron microscope (SEM, ZEISS EVOMA10), with 15 kV electron accelerating voltage, and then an EDS system attached at the SEM was used to analyze the elemental composition of point components on the sample surface.

### 2.4. Electrochemical Measurements

An electrochemical workstation (chi660e, CH Instruments Ins., Shanghai, China) was used to test the polarization curve and electrochemical impedance spectroscopy (EIS) of the samples by a three-electrode system, which was assembled by an experimental sample, platinum sheet, and saturated calomel electrode, as the working electrode, counter electrode, and reference electrode, respectively. The exposed areas of both working and counter electrodes were fixed at 1 cm$^2$. The electrolyte was 3.5 wt.% NaCl solution, and its temperature was around 25 ± 1 °C. The polarization curves were obtained by potentiodynamic scanning ranging from OCP ± 0.5 V (vs. SCE) at a scan rate of 5 mVs$^{-1}$. The EIS were acquired at opening circuit potentials (OCPs) with the amplitude of 10 mV in a scanning frequency ranging from 1 MHz to 0.01 Hz; then, the obtained data were fitted into EIS curves via the Zview software. The electrochemical samples were polished with a 2000 grade SiC sandpaper.

### 2.5. Mg–Air Batteries Performance Test

A self-designed battery test system was used. Its schematic diagram and actual image are shown in Figure 1. The cathode was composed of a catalytic layer, collecting layer, and waterproof breathable layer, which was made by a commercial C/MnO$_2$, nickel mesh, and PTFE particles, respectively. The electrolyte was a 3.5 wt.% NaCl aqueous solution. The reaction area of both anode and cathode was 2 cm$^2$. The curve of battery voltage vs. current density was recorded by the battery test system. The Mg–air battery was discharged for 12 h at a current density of 2.5 mA cm$^{-2}$ and 5 h at a current density of 30 mA cm$^{-2}$. The discharge products were removed by chromic acid (200 g L$^{-1}$ CrO$_3$ + 10 g L$^{-1}$ AgNO$_3$). Scanning electron microscopy (SEM) was used to observe the morphologies after discharging. The anode utilization efficiency and specific capacity were calculated by the mass loss method. The corresponding equations are as follows [12,14]:

$$\text{anode utilization efficiency (\%)} = \frac{M_t}{M_a} \times 100\% \tag{1}$$

$$\text{node specific capacity } \left(\text{mAhg}^{-1}\right) = \frac{I \times t}{M_a} \times 1000 \tag{2}$$

where $M_t$ is the theoretical mass loss (g) at a certain discharge current, and $M_a$ is the actual mass loss (g) after discharge. The theoretical mass loss was calculated by the equation [15].

$$M_t = \frac{I \times t}{F \times \Sigma \left(\frac{x_i \times n_i}{m_i}\right)} \tag{3}$$

where $I$ is the impressed current (A); $t$ is the discharge time (s); $F$ is the Faraday constant (96,485 C·mol$^{-1}$); $x_i$, $n_i$, and $m_i$ are the mass fraction, ionic valence, and molar mass (g·mol$^{-1}$) of an alloying element, respectively.

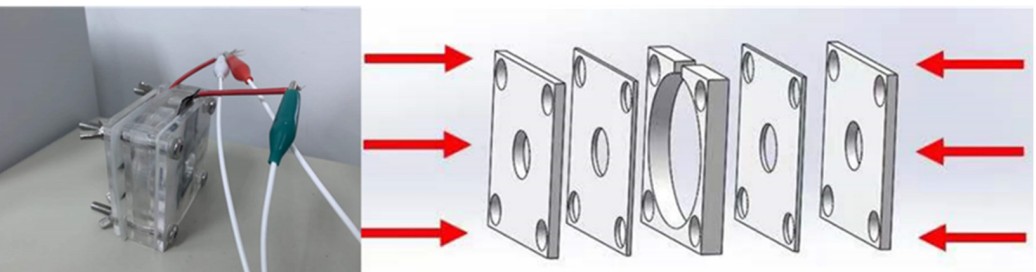

**Figure 1.** Actual image and schematic diagram of homemade Mg–air primary battery.

## 3. Results and Discussion

### 3.1. Microstructures

Figure 2 shows the optical microscopy (OM) images of as-cast and extruded Mg-In-Ca alloys with different amounts of Ca content. As can be inferred from Figure 2a–e, the microstructures of as-cast Mg-In-Ca alloy are mainly dendrite crystals. The dendrite crystals are coarse and developed with a low amount of Ca, as shown in Figure 2a. Nevertheless, the refinement effect of Ca can be proven by increasing the content. Moreover, the grain boundary becomes much clearer when using a high amount of Ca than that observed using other concentration levels. The formation of dendrites is due to the significant structural supercooling near the solid–liquid interface caused by high Ca content ($\geq$1%) [16], which improves the instability of the interface. Composition overcooling is also the main driving force for nucleation. In the composition overcooling region [17], abundant nucleation sites can be activated to form a more crystal nucleus. From the scanning electron microscopy (SEM) images in Figure 3a–e and the EDS spectrum results, we can conclude that the eutectic organization appears, which is composed of $\alpha$-Mg matrix and Mg$_2$Ca phase in the as-cast Mg-In-Ca alloy. This is because the solid solubility of In in Mg is relatively high (53.2 wt.%), while Ca has almost no solubility (1.35 wt.%), which induces the formation of intermetallic compounds. During solidification, eutectic phases are also consequently formed. The eutectic phases are distributed in grain boundaries, which seems like a semi-continuous network structure in the beginning. With increasing Ca content, the eutectic phases are interconnected, forming a continuous network structure, and the average thickness of the eutectic phases becomes larger as Ca content increases. The grains are refined, caused by the hindrance effect of large eutectics on grain growth. Another aspect to consider is that due to the low solubility of Ca in Mg, a large number of Ca atoms are gathered near the solid–liquid interface, and the slow diffusion of Ca atoms limits the growth of grains [16–18]. For Mg-In-Ca alloy with high Ca content, the eutectic structure is composed of an $\alpha$-Mg matrix, and the Mg$_2$Ca phase exists both along the grain boundary and in the interdendritic region. Obviously, it can be inferred from SEM images and other papers [16,17,19] that the increase in Ca content also leads to a significant increase in the Mg$_2$Ca phase.

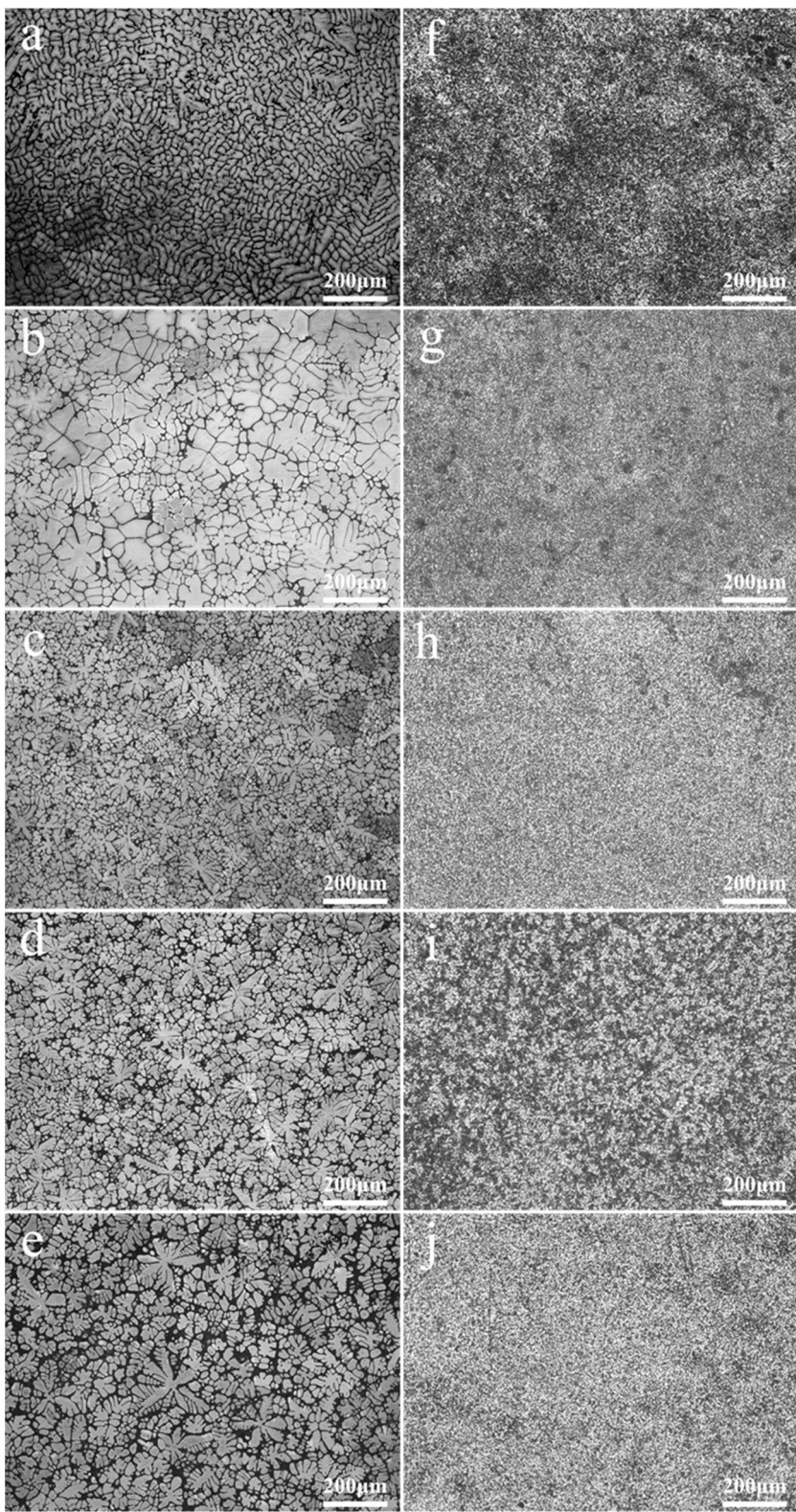

**Figure 2.** The optical metallographic microstructures of Mg-In-Ca alloys: (**a**) as-cast IC31 alloy, (**b**) as-cast IC32 alloy, (**c**) as-cast IC33 alloy, (**d**) as-cast IC34 alloy, (**e**) as-cast IC35 alloy, (**f**) extruded IC31 alloy, (**g**) extruded IC32 alloy, (**h**) extruded IC33 alloy, (**i**) extruded IC34 alloy, and (**j**) extruded IC35 alloy.

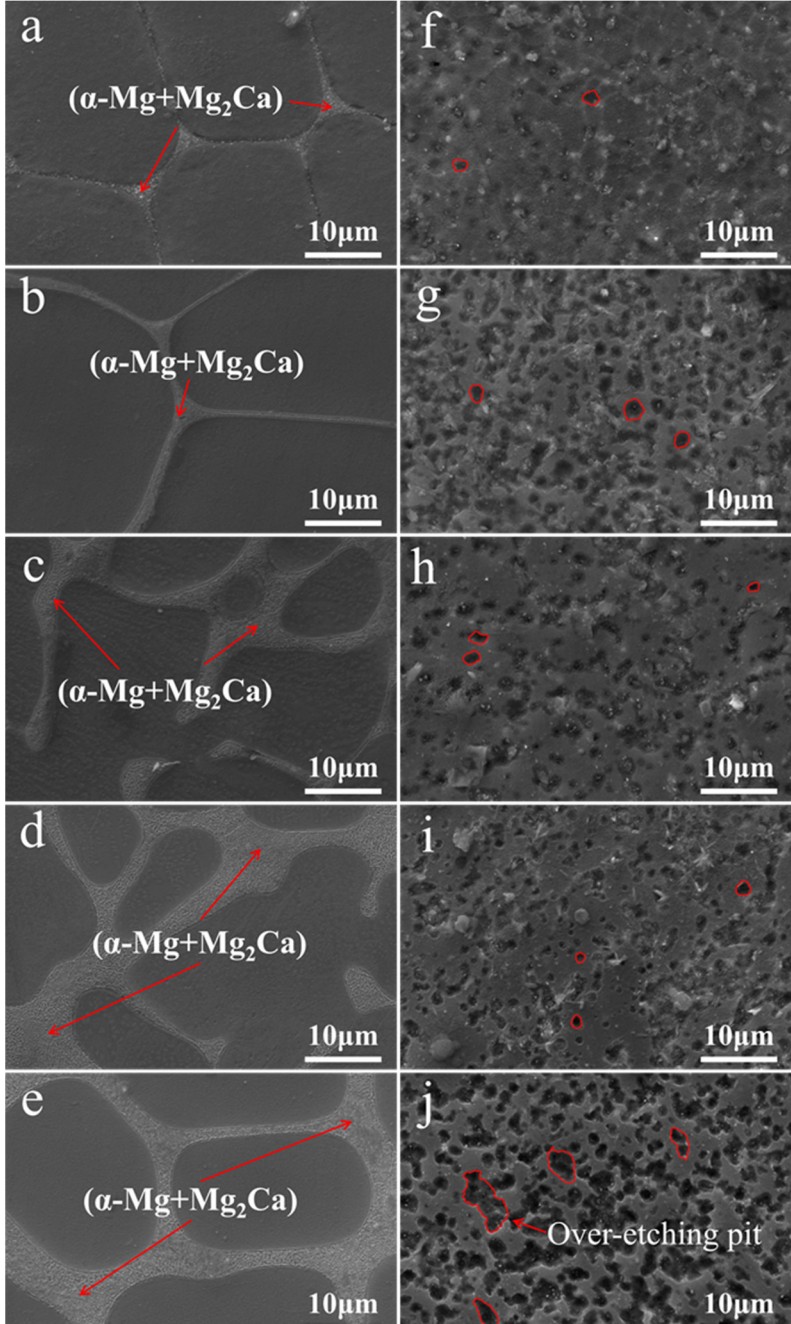

**Figure 3.** SEM images of Mg-In-Ca alloys: (**a**) as-cast IC31 alloy, (**b**) as-cast IC32 alloy, (**c**) as-cast IC33 alloy, (**d**) as-cast IC34 alloy, (**e**) as-cast IC35 alloy, (**f**) extruded IC31 alloy, (**g**) extruded IC32 alloy, (**h**) extruded IC33 alloy, (**i**) extruded IC34 alloy, and (**j**) extruded IC35 alloy.

As can be seen from Figure 2f,g, after extrusion, the $Mg_2Ca$ phases are crushed, and the grain boundaries are therefore not so clear as the as-cast samples. Normally, the broken $Mg_2Ca$ phases effectively promote recrystallization, contributing to grain refinement. Furthermore, the morphology distribution of $Mg_2Ca$ phases can be observed from Figure 3f,g. $Mg_2Ca$ phases are dispersed in the Mg matrix, and the number of $Mg_2Ca$ phases increases with increasing Ca content. For samples with high Ca content, such as IC34 and IC35, their $Mg_2Ca$ phases are interconnected with each other, forming fiber-like morphologies, which means the eutectic phases of these alloys are difficult to be completely refined by the extrusion process.

*3.2. Mg–Air Battery Performance*

Figure 4 is the galvanostatic discharge curve of pure Mg and Mg-In-Ca alloys as anodes of the Mg–air battery at a current density of 2.5 mA·cm$^{-2}$ and 30 mA·cm$^{-2}$. It can be seen from Figure 4a that the voltage levels of pure Mg, as-cast, and extruded Mg-In-Ca alloy anodes basically reach stable states after discharging for 2500 s, indicating that the dissolution and self-corrosion of the Mg anodes achieve dynamic balance. The discharge performance of Mg–air batteries is statistically listed in Table 2. Battery with pure Mg anode shows the lowest average voltage among both as-cast and extruded IC batteries. Through comparison, alloying element In obviously increases the discharge voltage, which means In activates the Mg anode. IC30 acquires the highest value, reaching 1.41 V. On the contrary, adding Ca into Mg-In alloys lowers their voltage. The difference in IC31 to IC35 on discharge voltage is inconspicuous, only 0.03 V (range from 1.30 V to 1.33 V). A similar tendency can be observed in high discharge current density (Figure 4b). However, as current density increases to 30 mA cm$^{-2}$, the discharge curves become fluctuant with slightly rising. The discharge reactions are increasingly difficult to be controlled. Among the alloys, without adding In or Ca, pure Mg shows a larger amplitude of fluctuation than IC series Mg alloys.

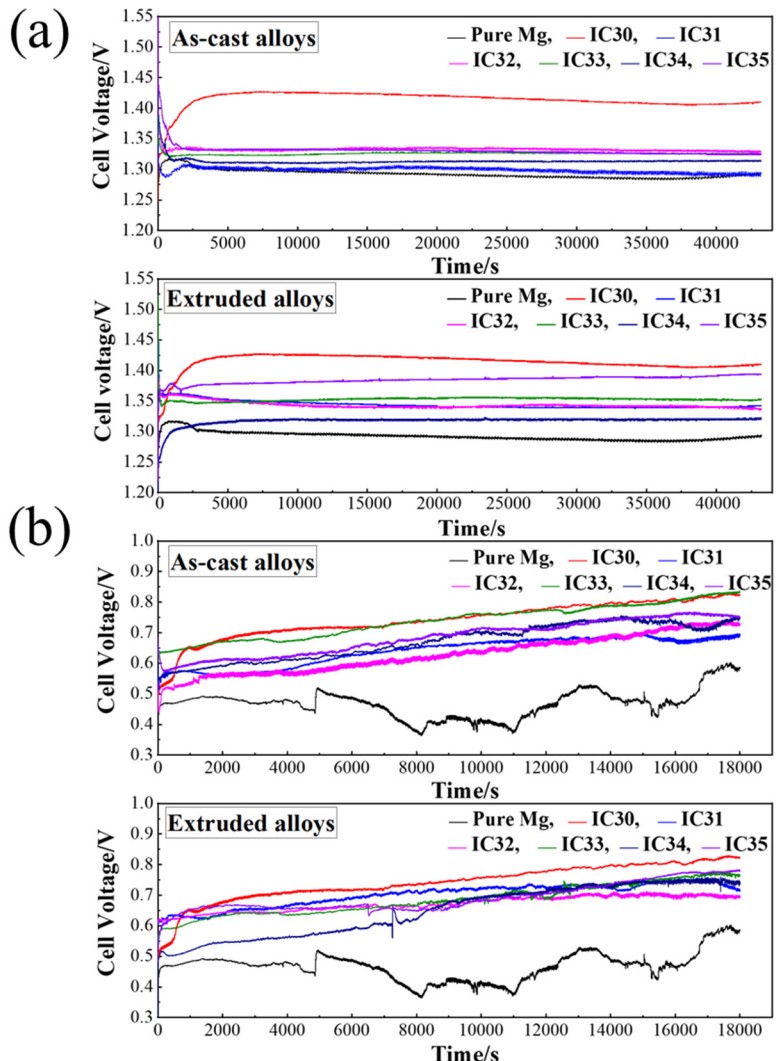

**Figure 4.** Galvanostatic discharge curve of pure Mg and Mg-In-Ca alloys at (**a**) 2.5 mA·cm$^{-2}$ discharge current for 12 h and (**b**) 30 mA·cm$^{-2}$ discharge current for 5 h.

**Table 2.** Discharge parameters of Mg–air batteries with different alloy anodes at different current densities.

|  | Samples | Current Density/ mA cm$^{-2}$ | Discharge Time/ h | Average Voltage/ V | Specific Capacity/ mAh g$^{-1}$ | Anodic Efficiency/ % |
|---|---|---|---|---|---|---|
| as-cast | Pure Mg | 2.5 | 12 | 1.29 | 584.80 | 26.18 |
|  | IC30 | 2.5 | 12 | 1.41 | 270.15 | 12.37 |
|  | IC31 | 2.5 | 12 | 1.30 | 478.47 | 22.07 |
|  | IC32 | 2.5 | 12 | 1.33 | 698.49 | 32.29 |
|  | IC33 | 2.5 | 12 | 1.33 | 813.00 | 37.91 |
|  | IC34 | 2.5 | 12 | 1.31 | 781.25 | 37.00 |
|  | IC35 | 2.5 | 12 | 1.33 | 483.48 | 22.87 |
|  | Pure Mg | 30 | 5 | 0.468 | 887.57 | 39.74 |
|  | IC30 | 30 | 5 | 0.742 | 1007.7 | 46.14 |
|  | IC31 | 30 | 5 | 0.64 | 724.46 | 33.41 |
|  | IC32 | 30 | 5 | 0.629 | 1294.22 | 59.84 |
|  | IC33 | 30 | 5 | 0.738 | 1323.92 | 61.74 |
|  | IC34 | 30 | 5 | 0.670 | 1238.65 | 58.64 |
|  | IC35 | 30 | 5 | 0.684 | 655.88 | 31.02 |
| extruded | IC31 | 2.5 | 12 | 1.34 | 639.66 | 29.50 |
|  | IC32 | 2.5 | 12 | 1.34 | 672.65 | 31.10 |
|  | IC33 | 2.5 | 12 | 1.35 | 698.49 | 32.57 |
|  | IC34 | 2.5 | 12 | 1.32 | 603.62 | 28.58 |
|  | IC35 | 2.5 | 12 | 1.38 | 554.53 | 26.23 |
|  | IC31 | 30 | 5 | 0.701 | 836.35 | 38.57 |
|  | IC32 | 30 | 5 | 0.672 | 1248.44 | 57.72 |
|  | IC33 | 30 | 5 | 0.685 | 1118.15 | 52.14 |
|  | IC34 | 30 | 5 | 0.649 | 1129.94 | 53.50 |
|  | IC35 | 30 | 5 | 0.691 | 1008.06 | 47.68 |

To explain the above phenomenon, firstly, the effect of element In should be discussed. During the discharge process of the Mg matrix, In atoms are oxidized at the same time to produce $In^{3+}$. Since $Mg$ have more negative electrode than $In$, $In^{3+}$ reacts with the Mg matrix and redeposits at the interface between the anode and the discharge product layer, which reduce the adhesion of discharge products, making the discharge product layer easy to crack and fall off from the anode surface. Electrolyte penetrates cracks and continuously contacts anode to react, which weakens the passivation effect of the oxide layer and improves the discharge voltage of the battery. This phenomenon is further confirmed by the SEM images in the next section. The displacement reaction of redeposition of In on the electrode surface can be expressed by Equation (4) [20] as follows:

$$3Mg(s) + 2In^{3+}(aq) = 3Mg^{2+}(aq) + 2In(s) \tag{4}$$

Secondly, the element Ca somehow passivates the Mg anode. Ca in Mg alloy mainly exists in the form of the second phase $Mg_2Ca$. $Mg_2Ca$ phases are located in grain boundaries, acting as barriers. A galvanic couple is formed by the Mg matrix and $Mg_2Ca$ phase, which promotes both discharge and self-corrosion reactions. Moreover, the interconnected $Mg_2Ca$ barriers (e.g., IC34, IC35 in Figure 3) accelerate Mg dissolution to generate a large number of discharge products at the interface between anode and electrolyte, thus lowering the discharge voltage. Nevertheless, fortunately, Ca refines the grain size of Mg, which is demonstrated in Figure 2. Reducing grain size helps to enhance the utilization efficiency of the anode. Therefore, although the discharge voltage of IC33 is lower than IC30, its anode utilization and specific capacity exhibit maximum level reach 61.71% and 1323.92 mAh g$^{-1}$ at 30 mA cm$^{-2}$ current density, respectively. For pure Mg and IC30, the corresponding values are only 39.74%, 887.57 mAh g$^{-1}$, and 46.14, 1007.7 mAh g$^{-1}$. It can be concluded that the composition of Ca should be optimized, in order to generate appropriate synergistic effects.

### 3.3. Surface Morphologies during Discharging

To explore the discharge status of IC Mg alloys, SEM and EDS analyses were carried out. Figures 5 and 6, respectively, show the morphologies of IC alloy anodes before and after removing discharge products. All images were captured after a long period of discharging in a 3.5% NaCl solution. As can be seen from Figure 5, the product morphologies of IC33 at low current density are significantly different from that those at high current density. At high current density, product layers (Figure 5c,d) show more cracks than their counterparts, even forming some deep grooves. Groove allows the electrolyte to fully contact the electrode, thus enhancing the discharge activity of the electrode [21]. Due to the effects of In, such morphologies allow $Cl^-$ to adsorb on the anode surfaces and promote the compatibility of electrodes and electrolyte solution [22]. As evident from the EDS results shown in Figure 6b–d, In content on the electrode surface after discharging is much higher than that on the original alloy anode, indicating that In is redeposited on the electrode surface. In deposits reduce the adsorption force of discharge products on the anode surface, accelerating the falling off of discharge products and forming a loose discharge product layer. The loose discharge product layer with numerous cracks increases the effective reaction area of the electrode, thereby enhancing the electrochemical activity of the electrode [1,23]. Therefore, Mg-In-based anodes acquire strong discharge activity.

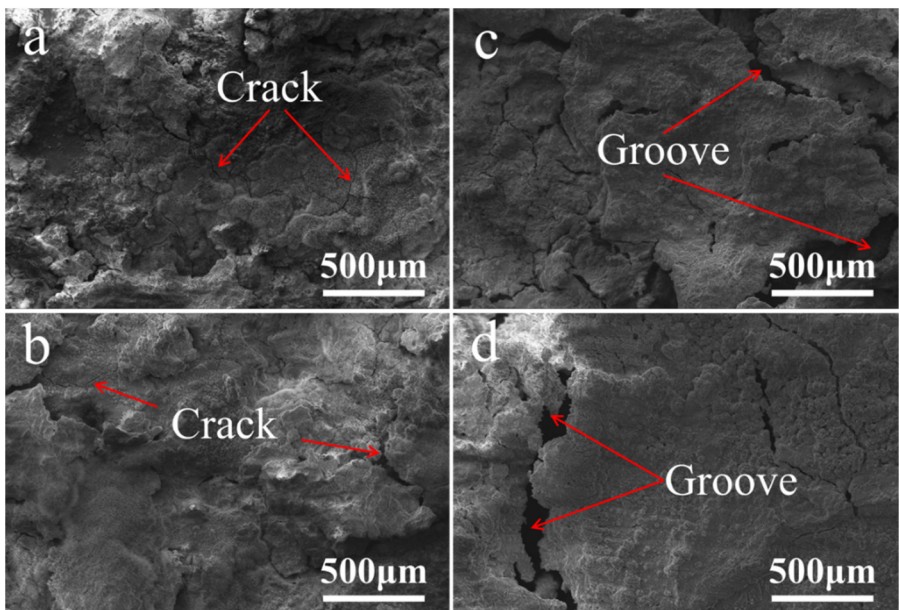

**Figure 5.** The morphologies of discharge products on IC33 alloy during discharging in 3.5% NaCl solution at different current densities: (**a**) as-cast, 2.5 mA cm$^{-2}$; (**b**) extruded, 2.5 mA cm$^{-2}$; (**c**) as-cast, 30 mA cm$^{-2}$; (**d**) extruded, 30 mA cm$^{-2}$.

Figure 6 shows the corrosion morphologies of IC31, IC33, and IC35 anodes after discharging for 5 h at a current density of 30 mA cm$^{-2}$. The oxide layers were removed in advance. The corrosion morphologies between extruded and as-cast anodes are obviously different. As can be seen from Figure 6a–c, it seems the grain boundaries preferentially react with the electrolyte. Furthermore, grain shedding can be found on the surfaces of the as-cast IC31 alloy anode and as-cast IC35 alloy anode, as shown in the red frame. This is mainly because $Mg_2Ca$ phases exist in grain boundaries during the discharging process, and therefore, the dissolution of $Mg_2Ca$ phases takes priority. As the discharging process advances, the grain boundary collapses. The adhesion of isolated grain becomes weaker and weaker, and finally, the undissolved grain falls off from the anode. In other words, the nonuniformity of reaction results in a reduction in anode efficiency. The anode surface of as-cast IC33 alloy is most uniformly dissolved, and little grain shedding can be detected. As-cast IC33 alloy anode has the highest anode utilization efficiency and specific capacity,

which is about two times (61.74%, 1323.92 mAh g$^{-1}$) larger than IC31 and IC35 (33.41%, 724.46 mAh g$^{-1}$ and 31.02%, 655.88 mAh g$^{-1}$).

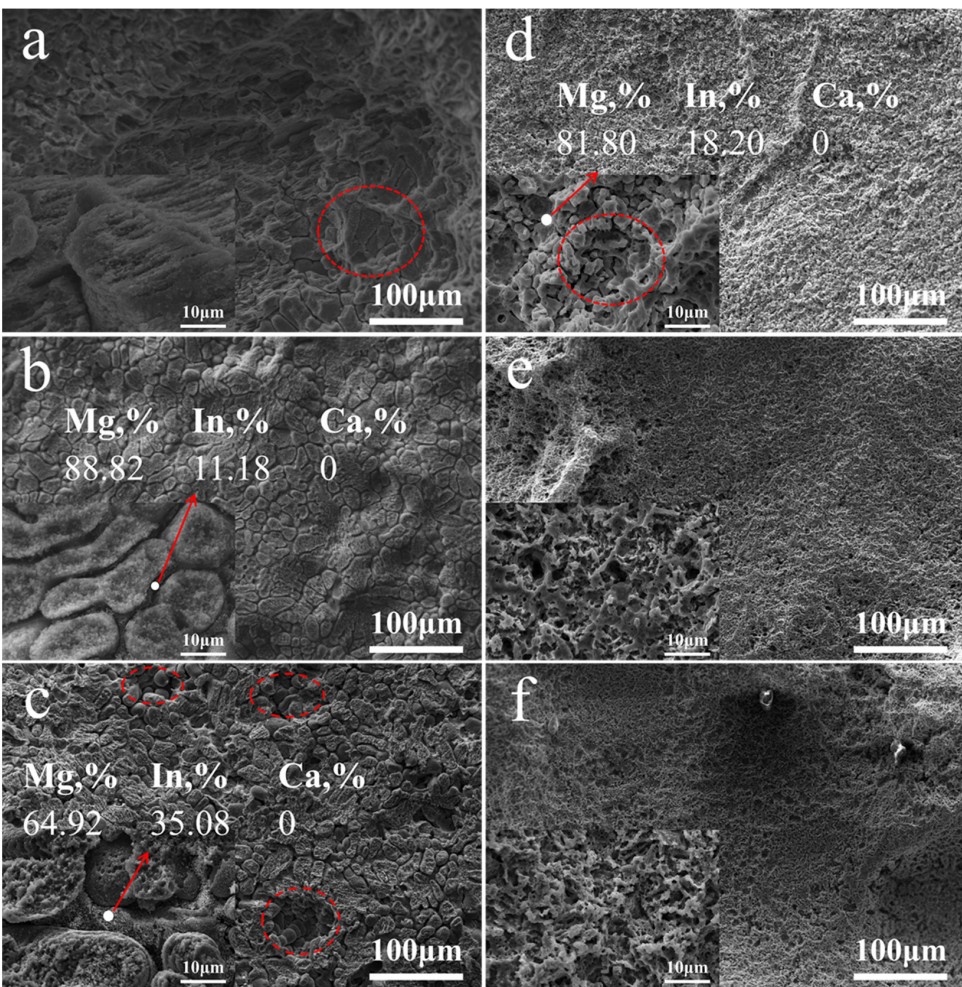

**Figure 6.** Corrosion morphologies of Mg-In-Ca alloy discharged in 3.5% NaCl solution for 5 h at current density of 30 mA cm$^{-2}$: (**a**) as-cast IC31 alloy; (**b**) as-cast IC33 alloy; (**c**) as-cast IC35 alloy; (**d**) extruded IC31 alloy; (**e**) extruded IC33 alloy; (**f**) extruded IC35 alloy.

As for the extruded alloy anodes, we can also find grain shedding on the surface. Although grain shedding greatly decreases the capacity, compared with the as-cast IC31 and IC35 anodes, the anode capacities and the average discharge voltages of extruded ones are slightly improved. It should be noted that the advantage of the extrusion process is attributed to the grain refinement effect. The anodes with refined grain size and expanded grain boundary possess large reactive sites, which ensure each small grain dissolves sufficiently.

The effects of adding elements In and Ca into Mg anode were investigated. Figure 7 shows morphologies of discharge products on pure Mg, IC30, and IC33 alloys after discharge. A comparison of these alloys reveals that pure Mg has the most compact and uniform discharge layer, thereby hindering both corrosion and continuous discharge reaction. The discharge layer on IC30 critically cracks, allowing more electrolytes to penetrate and react with the anode. However, in such cases, electrolytes could cause serious corrosion as well. Apparently, adding Ca relieves crack to a certain extent, shown as IC33 in Figure 7c. In fact, element Ca protects the anode from being corroded and guarantees the anode discharge with high anodic efficiency and specific capacity, as listed in Table 2. However, high Ca results in a large number of Mg$_2$Ca phases, which dissolve preferentially and trigger grain shedding.

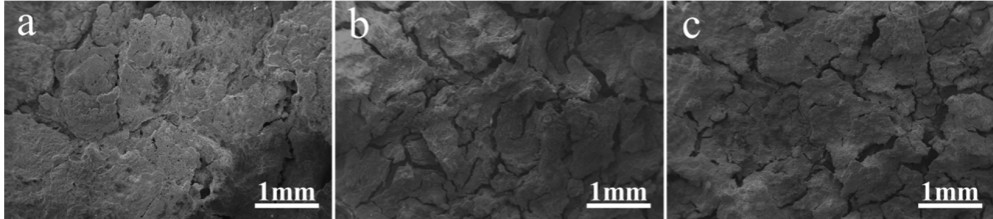

**Figure 7.** The morphologies of discharge products on (**a**) pure Mg, (**b**) IC30, and (**c**) IC33 during discharging in 3.5% NaCl solution at 30 mA cm$^{-2}$.

*3.4. Electrochemical Analysis*

Figure 8 shows the polarization curves of pure Mg, Mg-In alloy, and Mg-In-Ca alloy in 3.5% NaCl solutions. The detailed electrochemical parameters obtained from the polarization curves are summarized in Table 3. The polarization curve is composed of cathode branch and anode branch, which are related to hydrogen evolution and electrode oxidation, respectively [16,24]. The cathodic slopes (b$_c$) of pure Mg and IC33 alloys are obviously different, indicating that they show different hydrogen evolution behavior in the process of cathodic polarization. This is because the extruded alloy has a uniform microstructure, as well as due to the existence of a second phase. The smaller the b$_a$ of the anode, the stronger the discharge activity of the anode. The b$_a$ of the IC30 anode is the smallest, indicating that the discharge activity of the IC30 anode is the strongest. In the previous full battery discharge, the discharge voltage of the IC30 anode is the highest, regardless of whether the current density is high or low. The two are consistent. This is due to the activation of In, which can promote the adsorption of Cl$^-$ on the anode surface and destroy the passive film on the anode surface. The b$_a$ of IC33 alloy is the largest, which is attributed to the addition of Ca. However, the Mg$_2$Ca phase can reduce the polarization resistance (R$_P$). The R$_P$ of the IC33 anode is only 154.6 $\Omega$ cm$^2$, which is much smaller than that of pure Mg and IC30 anode. Although the discharge voltage of the IC33 anode is lower than that of the IC30 anode, it is also higher than that of pure Mg.

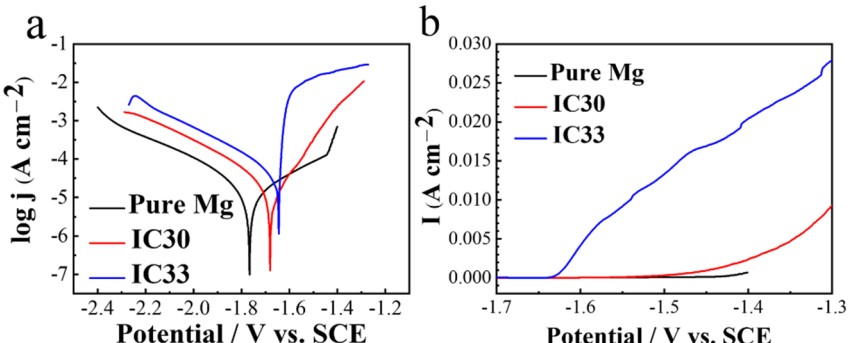

**Figure 8.** (**a**) Polarization curves and (**b**) anodic branches of the polarization curves for Mg-based anodes in 3.5% NaCl solution.

**Table 3.** Relevant corrosion parameters of Mg-In-Ca alloy.

| Samples | E$_{corr}$ (vs. SCE)/ V | J$_{corr}$/ mA cm$^{-2}$ | b$_a$/ mV dec$^{-1}$ | b$_c$/ mV dec$^{-1}$ | R$_P$/ $\Omega$ cm$^2$ |
|---|---|---|---|---|---|
| Pure Mg | −1.766 | 0.01018 | 212.68 | −162.23 | 3932.3 |
| IC30 | −1.681 | 0.01438 | 124.12 | −174.34 | 2192.5 |
| IC33 | −1.644 | 0.2960 | 234.80 | −190.80 | 154.6 |

As is revealed by corrosion parameters in Table 3, pure Mg has the most negative E$_{corr}$ and lowest J$_{corr}$, which means pure Mg has higher electrochemical activity under static conditions, and its parasitic reaction is restrained. However, pure Mg performs a

drastic anodic polarization. Although the existence of the $Mg_2Ca$ phase accelerates the corrosion of Mg alloy, IC33 represents low dissolve reaction overpotential in the anode process. For instance, if the discharge current density is fixed at 1 mA cm$^{-2}$ (the dashed line in Figure 8a), the potential of IC33 ($-1.621$ V) is much more negative than the other two, which implies IC33 is strongly activated in the discharge process. As can be more clearly seen from Figure 8b, IC33 acquires enhanced anodic kinetics, while pure Mg suffers serious polarization.

### 3.5. Electrochemical Impedance Spectroscopy (EIS)

The electrochemical behaviors of adding In and Ca were also investigated by electrochemical impedance spectroscopy (EIS) [25]. As shown by Figure 9a, each spectrum can be divided into three parts, including the capacitive loop at high frequency (signed as I), capacitive loop at a middle frequency (II) and inductive loop at low frequency (III). As illustrated by Figure 9b, two interfaces are formed between anode and electrolyte. The oxide layer–electrolyte interface and anode–oxide layer interface are, respectively, related to loop I and loop II. The inductive loop is probably caused by non-stationarity during measurement, which is not a key behavior in the discharging process to be considered.

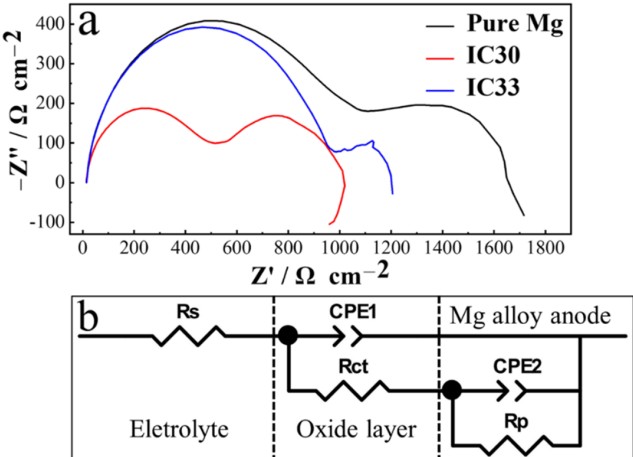

**Figure 9.** (**a**) Electrochemical impedance spectroscopy (EIS) of Mg-based anode in 3.5% NaCl solution; (**b**) equivalent circuit diagram.

Figure 9b shows the equivalent circuit around the surface of the anode during discharge. Calculated by the software "Zview", the values of simulated electronic components are listed in Table 4. $R_s$ is the resistance of the electrolyte, which is around 15 $\Omega$ cm$^2$ for each measuring system. As is well known, Mg anode can be inevitably passivated by the oxide layer during discharge. $R_f$ and $CPE_1$ are the resistance and capacitance of this oxide layer. Adding In obviously decreases the resistance of the oxide layer. The $R_f$ of IC30 (451.4 $\Omega$ cm$^2$) is two times lower than that of pure Mg (899 $\Omega$ cm$^2$), indicating that In has an influence on hindering the passivation of the oxide layer. This conclusion coincides with the observation of SEM in Figure 7. As regards the anode–oxide layer interface, $R_{ct}$ represents the resistance of charge transfer, while $CPE_2$ is the electric double layer capacitance on the surface of the anode. The charge transfer resistance reflects the activity of the electrode [26,27]. Generally speaking, the smaller the charge transfer resistance, the stronger the discharge activity of the Mg alloy anode. As can be seen from Table 4, In also reduces the $R_{ct}$ of pure Mg, from 801 $\Omega$ cm$^2$ to 532.4 $\Omega$ cm$^2$ (IC30). Meanwhile, IC33 has higher $R_f$ and lower $R_{ct}$ than IC30, which indicates that adding Ca induces forming compact oxide layer but dramatically reduces charge transfer resistance. With reference to the analysis of the polarization curve (Section 3.4), it can be inferred that $Mg_2Ca$ phases enhance anodic kinetics and promote the dissolving of the Mg matrix; thus, the anodic efficiency is further improved.

**Table 4.** EIS-simulated values of alloys.

| Samples | $R_s/$ $\Omega$ cm$^2$ | $Y_1/$ $\Omega$ cm$^{-2}$ sn | $n_1$ | $R_f/$ $\Omega$ cm$^2$ | $Y_2/$ $\Omega$ cm$^{-2}$ sn | $n_2$ | $R_{ct}/$ $\Omega$ cm$^2$ |
|---|---|---|---|---|---|---|---|
| Pure Mg | 15.16 | $1.53 \times 10^{-5}$ | 0.93 | 899.00 | $1.34 \times 10^{-3}$ | 0.59 | 801.00 |
| IC30 | 14.19 | $1.80 \times 10^{-5}$ | 0.92 | 451.40 | $1.79 \times 10^{-3}$ | 0.73 | 532.40 |
| IC33 | 14.84 | $2.11 \times 10^{-5}$ | 0.91 | 916.60 | $6.51 \times 10^{-3}$ | 0.72 | 284.10 |

*3.6. Discussion*

Table 5 shows the data comparison of different anodes after discharge. The applied current density of the anode in the discharge process is the same as that in this paper. When the applied current density is 2.5 mA cm$^{-2}$, AZ91-0.3La alloy has the highest discharge voltage. However, its specific capacity and anode efficiency are very low. Mg-3Al-1Ga alloy has the highest specific capacity and anode utilization, but its discharge voltage is very low. The IC33 alloy has balanced discharge activity. The discharge voltage of IC30 alloy is 1.41 V, which is only 0.04 V lower than the highest discharge voltage. When the applied current density is 30 mA cm$^{-2}$, the comprehensive discharge performance of Mg-6Al-1Sn alloy is the best. However, the anode efficiency of Mg-6Al-1Sn alloy is lower than that of IC33 alloy in this paper. The specific capacity of the two is not much different. Although the IC33 alloy in this paper is not the best, compared with the discharge properties of different alloys, IC33 alloy has balanced discharge properties and certain development potential. In the following experiments, the discharge performance of IC33 alloy was continuously improved by changing the hot extrusion temperature or adding other alloy elements.

**Table 5.** Electrochemical and battery properties of different alloys.

| Anode | Current Density/ mA cm$^{-2}$ | Average Voltage/ V | Specific Capacity/ mAh g$^{-1}$ | Anodic Efficiency/ % | Ref. |
|---|---|---|---|---|---|
| Pure Mg | 2.5 | 1.29 | 584.80 | 26.18 | - |
| IC30 | 2.5 | 1.41 | 270.15 | 12.37 | - |
| IC33 | 2.5 | 1.33 | 813.00 | 37.91 | - |
| Mg | 2.5 | 1.239 | 964 | 43.7 | [1] |
| Mg-3Al | 2.5 | 1.373 | 858 | 38 | [28] |
| Mg-3Al-1In | 2.5 | 1.415 | 943 | 42.2 | [28] |
| Mg-3Al-1Ga | 2.5 | 1.281 | 1185 | 53.1 | [28] |
| Mg-3Al-1Sn | 2.5 | 1.309 | 1162 | 52.1 | [28] |
| Mg-0.5Sn-0.5Mn-0.5Ca | 2.5 | 1.409 | 645 | 29.63 | [29] |
| Mg-6Zn-0.65Zr | 2.5 | 1.29 | 709 | 33.4 | [30] |
| Mg–6Zn–1Y | 2.5 | 1.38 | 828 | 39.2 | [30] |
| Mg–10Zn–1Y | 2.5 | 1.34 | 732 | 35.6 | [30] |
| AM60 | 2.5 | 1.148 | 872 | 39.5 | [12] |
| AZ91-1.5Ca | 2.5 | 1.41 | 711 | 31.7 | [31] |
| AZ91-0.5Sm | 2.5 | 1.44 | 614 | 17.6 | [31] |
| AZ91-0.3La | 2.5 | 1.45 | 520 | 23.1 | [31] |
| AZ91-1.5Ca-0.5Sm-0.3La | 2.5 | 1.43 | 543 | 24.5 | [31] |
| Pure Mg | 30 | 0.468 | 887.57 | 39.74 | - |
| IC30 | 30 | 0.742 | 1007.7 | 46.14 | - |
| IC33 | 30 | 0.738 | 1323.92 | 61.74 | - |
| Mg-3Al | 30 | 0.611 | 805 | 35.7 | [28] |
| Mg-3Al-1In | 30 | 0.617 | 903 | 40.5 | [28] |
| Mg-3Al-1Ga | 30 | 0.401 | 864 | 48.7 | [28] |
| Mg-3Al-1Ga | 30 | 0.494 | 949 | 42.5 | [28] |
| Mg-6Al-1Sn | 30 | 1.106 | 1339 | 59.1 | [32] |

## 4. Conclusions

In this research, a series of Mg-3In-xCa alloys as anodes of Mg–air batteries were successfully fabricated via casting and extrusion process. Afterward, the influences of adding In, Ca, and extrusion on microstructures and chemical components were analyzed by OM, SEM, and EDS. Mg–air primary cells were assembled to test the discharge performance. Moreover, we observed the surface morphologies during discharging and measured the electrochemical properties of Mg-based anodes. All of the analyses were investigated and discussed.

The addition of Ca element and extrusion process can effectively refine the grains. Ca exists in the form of the $Mg_2Ca$ phase, mainly distributed in grain boundary as a barrier, which promotes discharge performance. Compared with all the Mg-3In-xCa alloys, Mg-3In-3Ca alloy represents the best anodic performance, with 0.738 V of discharge voltage, 1323.92 mAh $g^{-1}$ of specific capacity, and 61.74% of anodic efficiency, owing to the synergistic effects of In and Ca on decreasing compactness of discharge products and enhancing anodic kinetics. On this basis, adding other elements to fabricate quaternary alloys with low corrosion is planned for further study.

**Author Contributions:** Conceptualization, G.Z. and H.L. (Huikun Liu); methodology, H.L. (Huikun Liu); software, J.W.; validation, H.Y., K.C. and Y.H.; formal analysis, G.Z.; investigation, H.L. (Huikun Liu); resources, D.X.; data curation, H.L. (Huikun Liu); writing—original draft preparation, G.Z. and H.L. (Huikun Liu); writing—review and editing, H.L. (Huikun Liu) and G.Z.; visualization, H.L. (Hang Li); supervision, J.Z.; project administration, S.T.; funding acquisition, G.Z. All authors have read and agreed to the published version of the manuscript.

**Funding:** This research was funded by the Key Research and Development Plan of Shandong Province, China, Grant Number 2019GGX102047. Natural Science Foundation of Shandong Province, China, Grant Number ZR2020QE021, ZR2020QE025. Youth Fund of Qilu University of Technology (Shandong Academy of Sciences), China, Grant Number 2020QN0021. National Natural Science Foundation of China, China, Grant Number 51901117.

**Institutional Review Board Statement:** Not applicable.

**Informed Consent Statement:** Not applicable.

**Data Availability Statement:** Not applicable.

**Conflicts of Interest:** The authors declare no conflict of interest.

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
