# Peer review of "The Discharge Performance of Mg-3In-xCa Alloy Anodes for Mg–Air Batteries"

_coatings, doi:10.3390/coatings12040428_

Round 1

Reviewer 1 Report

The title of this study is: The Discharge Performance of Mg-3In-xCa Alloy Anodes for Mg-air Battery.

I commented on the manuscript and the comments are presented below:

Part 1: Introduction.

The Introduction to the study is brief and does end with a clearly stated purpose or goals that the Authors wish to pursue.

Part 2: Material and Methods

The Methods section provides the reader with not enough information to repeat the experiments conducted. No methods of statistical analysis of the obtained results were used.  What were the parameters used in the microscopic examination? What methods were used to analyze the obtained images?

Part: 3 Results

For the most part the Results section is well structured.

Part: 4 Discussion

In the Discussion chapter, there is no full comparison and confrontation with the research of other authors in this area. The results were not fully discussed. A full discussion of the results obtained with other work in this field should be carried out in more aspects. I suggest supplementing the Chapter with additional information.

Part: 5 Conclusion

The Conclusions chapter contains information obtained after conducting experiments but performing only base statistical analyzes and were no comparison and confrontation with the research of other authors in this area.

Part: References.

The literature used is appropriate but should be supplementing about the items from the last years of publication about similar problem. The literature should be supplemented with additional items describing the examined aspects.

Reviewer 2 Report

The authors have shown the synthesis of Mg-3In-xCa Alloy and further used it as an anode for Mg-air Battery. The authors have tried to present some good results, however, there are some issues and suggestions that need to be addressed before publication in Coatings Journal.

  1. I simply don’t understand how to write Mg-3In-xCa correctly, write it correctly throughout with proper subscripts and superscripts.
  2. 3 keywords are very few, write at least 5 important keywords.
  3. The degree sign is used differently in different places. Please keep it uniform.
  4. There are a lot of grammatical and typo mistakes in the paper. The presentation is also very okay. Please improve the quality of the figures.
  5. Maintain the uniformity write either Fig. or Figure throughout.
  6. In Figure 2, SEM images are not clear.
  7. The electrochemical explanation is very weak throughout the manuscript and has weak alignment with material characterization results. Please rewrite and elaborate on the electrochemical section of the Mg-air Battery with appropriate explanations.
  8. It is highly suggested to make a comparative table to compare the performance of Mg-3In-xCa Alloy with other related literature in context to Mg-air Battery.
  9. 10 Figures are too much for the main manuscript. Either adjust them with each other or move to support information.
  10. Please give TEM and XPS for Mg-3In-xCa Alloy.

Round 2

Reviewer 1 Report

The authors referred to the comments from the previous review for the manuscript titled: The Discharge Performance of Mg-3In-xCa Alloy Anodes for Mg-air Battery. I accept explanations. In the future, I suggest using more precise  describing relationships between the parameters studied. They supplemented the discussion with a new literature data strengthens the message and importance of information in the manuscript.

Reviewer 2 Report

The manuscript can now be accepted in current form